

# Prevalence of permethrin-resistant kdr mutation in head lice (*Pediculus humanus capitis*) from elementary school students in Jeddah, Saudi Arabia

Isra M. Alsaady[1,2], Sarah Altwaim[2,3], Hattan S. Gattan[1,2], Maimonah Alghanmi[1,4], Ayat Zawawi[1,4], Hanadi Ahmedah[5], Majed H. Wakid[1,2] and Esam I. Azhar[1,2]

[1] Department of Medical Laboratory Sciences, Faculty of Applied Medical Sciences, King Abdul Aziz University, Jeddah, Saudi Arabia

[2] Special Infectious Agents Unit, King Fahad Medical Research Centre, King Abdul Aziz University, Jeddah, Saudi Arabia

[3] Department of Medical Microbiology and Parasitology, Faculty of Medicine, King Abdul Aziz University, Jeddah, Saudi Arabia

[4] Vaccines and Immunotherapy Unit, King Fahd Medical Research Center, King Abdul Aziz University, Jeddah, Saudi Arabia

[5] Department of Medical Laboratory Technology, Faculty of Applied Medical Sciences, King Abdul Aziz University, Rabigh, Saudi Arabia

Corresponding author
Isra M. Alsaady,
isra_alsaady@hotmail.com

## ABSTRACT

Head lice (*Pediculus humanus capitis*) are a major global concern, and there is growing evidence of an increase in head lice prevalence among Saudi schoolchildren. The purpose of this study is to investigate the prevalence of an insecticidal resistance mutation in head lice collected from schoolchildren. A polymerase chain reaction (PCR) was used to amplify a segment of the voltage-gated sodium channel gene subunit to assess the prevalence and distribution of the *kdr* T917I mutation in head lice. Subsequently, the restriction fragment length polymorphism (RFLP) patterns revealed two genotypic forms: homozygous-susceptible (SS) and homozygous-resistant (RR). The results showed that 17 (37.80%) of the 45 samples were SS, whereas 28 (62.2%) were RR and T917I and L920F point mutations were found in the nucleotide and amino acid sequences of RR. Compared to other nations, the frequency of permethrin resistance mutation in the head louse population in Saudi Arabia was low. This study provides the first evidence of permethrin resistance mutation in human head lice in Saudi Arabia. The findings of this study will highlight the rising incidence of the *kd*r mutation in head lice in Saudi Arabia.

## INTRODUCTION

One of the oldest ectoparasites known to infect humans worldwide is the louse (*Pediculus humanus*) (*Hard & Zinsser, 1935*). Two subspecies are well known to infest humans: body lice *Pediculus humanus humanus* and head lice *Pediculus humanus capitis*. The former have been infesting humans since prehistoric times, as their eggs have been discovered on the hairs of Egyptian mummies (*Hard & Zinsser, 1935*; *Sabbahy, 2017*). Head lice can

cause itching, leading to sleep deprivation, attention deficit, and subsequent secondary bacterial skin infections caused by rubbing an inflamed scalp. Unlike body lice, head lice do not transmit any disease except under experimental conditions (*Gratz, 1997*; *Sasaki et al., 2006*). They are more common in children (*Frankowski et al., 2010*) and are usually transmitted directly from head to head; thus, indirect transmission is less common (*Chunge et al., 1991*).

Head lice treatment is primarily based on physical removal (hair brushing or shaving) and pediculicides, which are applied topically to control the infestation. Commercially available pediculicides include natural pyrethrin esters (pyrethrum), synthetic pyrethroids (permethrin and phenothrin), organochlorine (lindane), organophosphates (malathion) and carbamate (carbaryl). The most common over-the-counter pediculicides are pyrethrins and synthetic pyrethroids, which were effective until the mid-1990s. Since then, many reports have described a reduction in their effectiveness (*Clark et al., 2013*).

Pyrethrins and pyrethroids have the same target site as DDT, the voltage-gated sodium channel (VGSC) on the neuron membrane. In the VGSC gene, three sodium channel mutations (M815I, T917I and L920F) have been related to permethrin resistance in head lice (*Lee et al., 2000*; *Lee et al., 2003*).in addition, the M815I and L920F mutations decrease the susceptibility of head lice to permethrin, while the T917I mutation is crucial in permethrin resistance and can be employed as a genetic biomarker for permethrin or pyrethroid resistance in head lice (*SupYoon et al., 2008*).

Several researchers have estimated the prevalence of head lice infestation in Saudi Arabia. For example, in Jeddah, *Boyle (1987)*) found that 12% of school students were infested with *Pediculosis humanus capitis*. Other research investigations in Saudi Arabia found a 5.2% infestation rate in Al-Khobar (*Al-Saeed et al., 2006*) and a 9.6% infestation rate in Abha (*Bahamdan et al., 1996*). Meanwhile, more recent studies have shown an increasing prevalence of infestation (45%) among female school pupils in Abha and Makkah (*Gharsan et al., 2016*; *Mohamed et al., 2018*). These results might indicate a reduction in pediculicide effectiveness.

Notably, their analysis did not include an examination of insecticidal resistance within head lice populations in this region.

Considering the above-mentioned studies, there is a lack of information regarding pyrethroid resistance in Saudi Arabia and the Middle Eastern area. A comprehensive investigation is necessary to gain insight into the present circumstances, prospects for potential outbreaks, as well as control methods. Thus, we aim to investigate the presence and prevalence of the *kdr* mutation (T917l, L920F) in head lice in Jeddah, Saudi Arabia.

## MATERIAL AND METHODS

### Head lice collection

Between October 2021 to February 2022, researchers performed a study in Jeddah intending to investigate the prevalence of head lice among young girls. A total of six primary schools located in diverse regions were selected for this purpose, each comprising around 600 students between the ages of 6–12 years. Ethical and regulatory compliance was ensured

by acquiring consent from parents and obtaining school agreements before initiating any screenings. To detect live head lice infestations, fine-toothed anti-louse combs were used under supervision by a qualified nurse within the school's premises. Outcomes revealed that children across all visited locations had varied levels (range 0% to 7%) of persistent head lice infections; only mature adult specimens accounted for these cases as evidenced through retrospective analysis after collecting one-five parasites from every student screened during examinations. After carefully examining each child's head, medical tweezers were used to collect the samples. The insects were placed in separate vials containing 95% ethyl alcohol and frozen at −20 C °.

Ethical permission was obtained from the ethical committee of the Faculty of Applied Medical Sciences, King Abdulaziz University (FAM EC2021-10). The participants of the study provided written consent following the guidelines.

## Genomic DNA extraction

Each louse's genomic DNA was collected using a procedure adapted from Toloza et al. (*Toloza et al., 2014*) First, all the lice were sliced in half and inserted into 1.5-ml Eppendorf centrifuge tubes containing cell lysis solution and proteinase K and then mashed with a plastic pestle. Subsequently, the DNA was extracted using the QIAamp DNA Mini Kit (Qiagen, Hilden, Germany) based on the manufacturer's instructions. To measure the amount of DNA in each sample, a NanoDrop 1000 spectrophotometer (Thermo Scientific, Waltham, MA, USA) was used, and the material was diluted to a 5–10 ng/mL concentration.

## Amplification of the VSSC gene

On the genomic DNA fragment of each louse, polymerase chain reaction (PCR) was employed to amplify a 332-bp fragment of the VSSC gene containing a region impacted by C/T mutation and the corresponding change in T917I amino acid (*Durand et al., 2007*). The reactions were performed in a 25-l reaction container containing 12.5 µL of PROMEGA MasterMix and 1µL (0.25 M) of each primer. 5′-AAATCGTGGCCAACGTTAAA-3′ (forward) and 5′-TGAATCCATTCACCGCATCC-3′ (reverse), 2 µL of DNA template and 8.5 µL of pure water.

The applied PCR conditions were as follows: 10 min at 94 °C, followed by 40 cycles of 94 °C for 30 s, 56 °C for 30 s, 65 °C for 1 min and a final extension step for 10 min at 65 °C. The C/T mutation was detected by digesting 10 l of each PCR product with the SspI restriction enzyme 10U (Thermo Scientific). Finally, this fragment was electrophoresed in a 2% agarose gel and observed with ethidium bromide under ultraviolet (UV) light.

## Screening of the *kdr* mutation using PCR–RFLP

The SspI restriction enzyme, which recognized the AAT|ATT restriction site, was used to identify the *kdr* T917I mutation linked to pyrethroid resistance in all collected samples. When the T917I amino acid changed, the restriction fragment length polymorphism (RFLP) pattern in the homozygous-resistant mutant (RR) genotype exhibited two fragments (*i.e.,* digestion). In the homozygous-susceptible or wild-type allele (SS) genotype, only one band of 332 bp was observed (*i.e.,* undigested). In the heterozygote (RS) genotype,

complete digestion resulted in two fragments (*i.e.,* 261 and 71 bp), while partial digestion resulted in three fragments (*i.e.,* 332, 261 and 71 bp).

In a final 10-l volume, the RFLP reaction mixture contained 500 ng of PCR products, 10X buffer G and 10U SspI restriction enzyme (Thermo Fisher Scientific, Waltham, MA, USA) nuclease-free water. The procedure began with a 90-min incubation at 37 °C, followed by 20-min heat inactivation at 65 °C. The digested products were separated on a 1% agarose gel electrophoresis at 100 V for 60 min and observed with ethidium bromide under UV light.

### Nucleotide sequencing

Direct DNA sequencing was perfomed on the PCR products to observe the homozygous susceptible and homozygous resistant genotype sequences. The purified PCR products by using MQ PCR/Gel product purification kit (MOLEQULE-ON) , were subjected to direct sequencing using forward and reverse primers specific for the VSSC gene, on the Illumina Hi-Seq 2500 (Illumina, San Diego, CA, USA).

### Statistical analysis

To estimate the frequencies of genotypes RR, RS, and SS by dividing the number of lice belonging to each genotype by the total number of analysed human head lice. Furthermore, they compared these frequencies against Hardy-Weinberg expectations and estimated Wright's inbreeding coefficient (*Weir & Cockerham, 1984*). The main goal was to assess any departure from Hardy-Weinberg proportions.

The analysis of the sequences was conducted using bioinformatics tools such as Basic Local Alignment Search Tool (BLAST; http://www.ncbi.nlm.nih.gov/blast/) FitchTV 1.4 and Clustal Omega. The resulting sequences were then aligned with representative wild type sequences of VSSC in GenBank using Molecular Evolutionary Genetics Analysis version 7.0 (MEGA) software.

## RESULTS

All 45 head lice collected were identified and tested for the *kdr* T917I substitution. After SspI digestion, the presence of one or two fragments is the specific genetic marker for permethrin resistance. In the *kdr* fragment, the CT nucleotide change, which codes for the T917I substitution, results in a unique restriction endonuclease cutting site. Consequently, the study revealed the existence of two types of head lice genotypes that were homozygous susceptible *kdr*-like alleles (SS); *kdr*-resistant homozygotes (RR); however, it was unexpected to find no heterozygotes (Fig. 1).

The *kdr* mutation was discovered in 62% of Jeddah head lice populations. The louse population had 17 (37.80 percent) homozygous susceptible *kdr*-like alleles (SS); 28 (62.2 percent) *kdr*-resistant homozygote (RR) was found in the populations studied (Table 1).

The Hardy–Weinberg (H-W) model was used to estimate the genotype frequency distribution of *kdr*. The chi-squared equation yielded a result of 2.68. Because this figure is smaller than the crucial value of 3.84, we cannot reject the null hypothesis that states substantial change in allele frequencies, and the *kdr* gene in this population is most likely at genetic equilibrium.

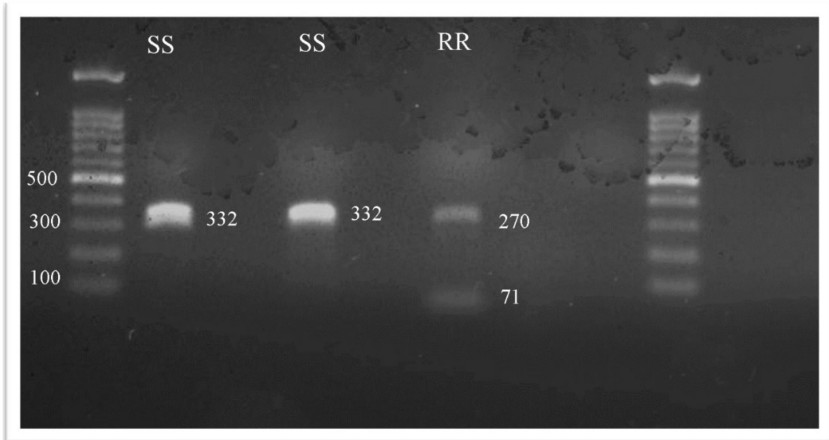

**Figure 1** The RFLP patterns of kdr T917I genotypes seen on a1% agarose gel electrophoresis.

**Table 1** T917I kdr-like allele distribution in Jeddah, Saudi Arabia head louse populations.

| Population | No. of head lice analyzed (no. of infested subjects) | Genotype | | Resistance allele frequency (%) | H-W[a] ($\chi^2$) | $F_{IS}$ |
|---|---|---|---|---|---|---|
| | | S/S | R/R | | | |
| Total | 45 | 17 | 28 | 62.2 | 2.68[b] | 1 |

1. A chi-square test was used to determine if populations were in the Hardy-Weinberg equilibrium. ($\chi^2 = 3.84$, $df = 1$, $P < 0.05$)
2. [b]Values that are statistically significant at $P < 0.05$. The significance level implies that the null hypothesis is not rejected.
3. $F$ is values > 0 indicate heterozygote deficiency, whereas Fis values < 0 indicate heterozygote excess

The frequency and genotype of head lice found in different areas of Jeddah city were studied; the highest frequency was discovered in south Jeddah (Fig. 2).

Using the ClustalW algorithm, the acquired nucleotide and amino acid sequences were aligned and compared to the wild-type sequence made available in the GenBank database. The reference sequence was a *Pediculus humanus capitis* variation that was insecticide-susceptible and had three amino acid changes at kdr mutation sites (accession no. AY191156) (Fig. 3). Multiple nucleotide and amino acid sequences were aligned, and it was found that homozygous resistant sequences had a T917I point mutation caused by a CT substitution, resulting in a Thr–Ile mutation, but homozygous susceptible sequences had no substitutions at codon 917. All homozygous resistant sequences furthermore revealed the L920F point mutation, which was brought on by a C–T nucleotide substitution and led to a Leu–Phe mutation.

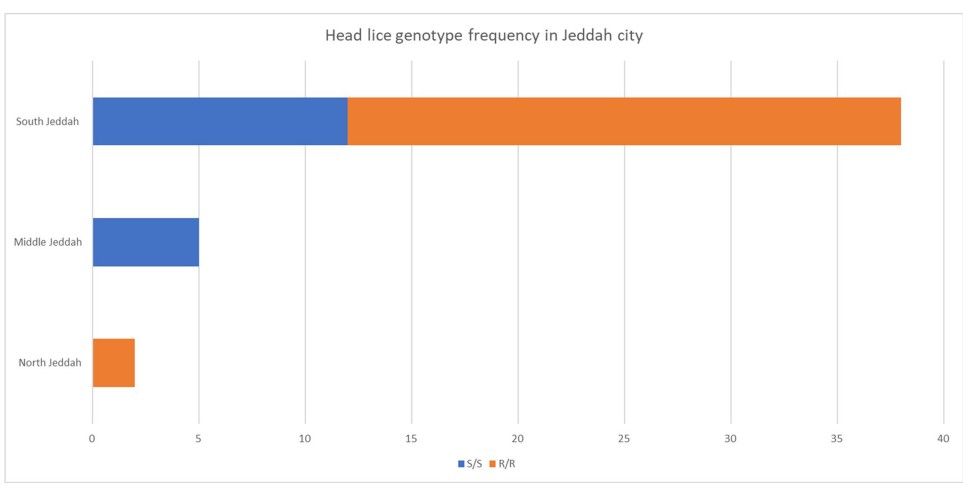

**Figure 2** **Head lice frequency in different areas of Jeddah city, Saudi Arabia.** S/S susceptible while R/R resistant.

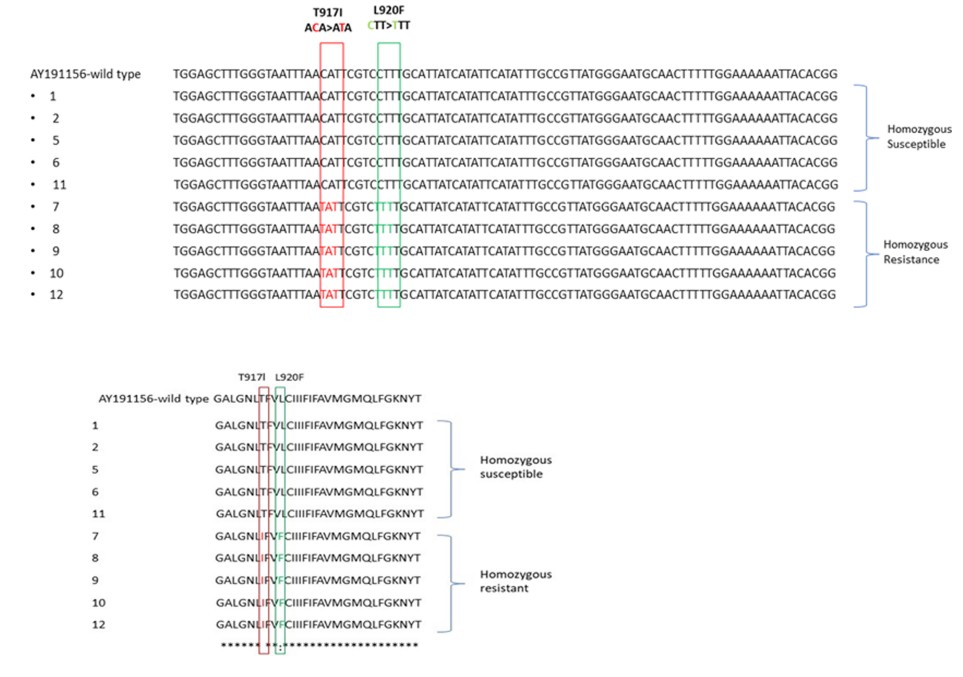

**Figure 3** **The alignment of nucleotide (A) and amino acid sequences (B) for the α subunit of the VSSC gene in head lice collected from Jeddah, Saudi Arabia is presented.** This sequence displays the location of two mutations: the kdr T917I mutation and the L920F mutations: the kdr T917I mutation and the L920F mutation, both marked by a vertical column.

## DISCUSSION

Head lice are found globally and are the most common ectoparasites that infect humans (*Feldmeier & Heukelbach, 2009*). They infest schoolchildren and are typically stressful for both the children and their parents. Furthermore, chemical pesticides, the most common of which is permethrin, are used as the first line of therapy for head lice. In the meanwhile, a significant factor in an increase in head lice infestations is drug resistance, which causes treatment to fail. Resistance is a permanent feature that an insect pest acquires over time as a result of selective pressure from frequent or insufficient pesticide use (*Durand et al., 2012*).

The *kdr*-resistant allele was identified in up to 98.3% of examined lice, and the T917I kdr mutation is one of three mutations discovered in human head lice from the United States, Canada, France, Argentina, Thailand, Honduras, and Chile. (*Toloza et al., 2014*; *Yoon et al., 2014*; *Eremeeva et al., 2017*; *Roca-Acevedo et al., 2019*; *Brownell et al., 2020*; *Larkin et al., 2020*).

This study considers the presence and distribution of the T917I mutation in the head louse population from Jeddah, Saudi Arabia. Forty-five head lice were collected and tested for the presence of the T917I *kdr* mutation. SS was found in 37% of the samples, and RR was found in 63% of the samples. A previous study from Saudi Arabia found that 69% of people did not respond to pyrethrin-containing pediculicide shampoo; these findings are similar to ours, though the non-responders included RS and SS genotypes (*Abdullah & Kaki, 2017*).

To validate the reliability of RFLP methods in identifying specific amino acid substitutions, samples were randomly selected from three genotypes for DNA sequencing. The results of the sequencing exhibited a direct correlation with the pattern of detected RFLP bands in two genotypes. In the chromatogram analysis, double peaks (representing two colors) were observed at T917I and L920F point mutations in homozygous genotype. Furthermore, distinct genomic DNA patterns for both wild-type and mutant sequences were generated on the chromatogram. These findings highlight PCR-RFLP as a promising technique for detecting knockdown resistance in head lice due to its accurate identification of specific point mutations through characteristic RFLP band patterns. .

*Brownell et al. (2020)* conducted a study in Thailand to investigate the presence of this mutation among primary school children and found similar results to ours, revealing that 60% of the distribution was SS, 22.31% was RS and 17.69% was RR. In addition, Larkin et al. investigated the mutation associated with pyrethroid resistance in head lice in Honduras and found that 6.1% of the distribution was SS, while 93.9% were RS. Meanwhile, RR was not detected in the studied population (*Larkin et al., 2020*). Furthermore, Roca-Acevedo et al. found that among the head louse population in Chile, 7% were SS, 88.8% were RS and 8.4% were RR (*Roca-Acevedo et al., 2019*). In 2014, Tolozo et al. estimated the resistance level among the head louse population in Argentina, and the results showed an increased level of pyrethroid-resistant *kdr* alleles varying between 67% and 100%. Of these, 85.1% were RR, 8.4% were SS and 6.5% were RS (*Toloza et al., 2014*). Moreover, a survey conducted among schoolchildren in France to evaluate permethrin resistance in the head
louse population revealed a high level of resistance in the study population (*Durand et al., 2012*).

Notably, the resistance mutation levels in these studies are much higher than those in this study, which could be attributed to the extensive usage of non-chemical therapies and the lack of pyrethrin and pyrethroids in most over-the-counter treatments (*Burkhart & Burkhart, 2000*).

Our results reveal a lack of heterozygotes (0 per 45) and the occurrence of homozygous *kdr*-resistant mutations in the examined populations in addition to the wild type. In contrast, previous studies reported only the incidence of homozygous *kdr*-resistant lice rather than heterozygotes (*Toloza et al., 2014*; *Yoon et al., 2014*; *Eremeeva et al., 2017*).

Thus, the absence of heterozygotes (0/45) in the studied populations, as well as the presence of homozygous *kdr*-resistant and homozygous-susceptible, indicate the existence of two distant populations. Furthermore, due to the absence of heterozygotes, genotype proportions deviated significantly from Hardy–Weinberg expectations. Several factors that induce heterozygote deficiency include the Wahlund effect, self-fertilization, and positive assortative mating. However, the positive $F_{IS}$ may suggest the Wahlund effect (*Waples & Allendorf, 2015*), which implies the presence of a subpopulation. Furthermore, the genetic diversity study of head lice in Saudi Arabia, revealed that the lice belong to two distinct clades. This phenomenon may be explained by the fact that Saudi Arabia is one of the most attractive countries for foreign labour and religious pilgrims from all over the world (*Al-Shahrani et al., 2017*).

However, our findings may offer a plausible explanation for the surge in Saudi Arabia's infestation rates over the past decade when contrasted with prior research studies conducted (*Gharsan et al., 2016*; *Mohamed et al., 2018*)due to head louse populations displaying mutations of insecticide resistance. Notably, our results show that school closures and adherence to social distancing guidelines implemented as a response measure against the COVID-19 crisis would significantly mitigate head lice infestations as expected by (*Galassi et al., 2021*).

In the future, monitoring and response to head lice infestation must improve, due to the high risk of resistance-type mutations like RR genotype leading to insecticide resistance and further outbreaks.

The mutation's geographical distribution is influenced by various factors (Fig. 2). This can be attributed to variations in the host population or differences in sample sizes collected from different locations. It is worth noting that the infection rate in the northern and central regions is considerably less than that of the southern region.

The limited sample size of this study was a result of parental reluctance to participate and the reduction in pest infestation during COVID-19 closure. To enhance comprehension about pyrethroid resistance in Saudi Arabia, a larger sample from diverse regions at varying time points would provide more substantial insights into genotype frequencies necessary for treatment and control selections. Improving the sample size could establish comprehensive support towards designing effective drug selection plans.

## CONCLUSION

This is the first study in Saudi Arabia to use a molecular approach to detect permethrin resistance-related mutation in human head lice. The PCR–RFLP method was applied to determine the presence of a *kdr* mutation in head lice, which revealed two different genotypes in Saudi Arabia. This information will highlight the presence of the *kdr* mutation in head lice and increase awareness of the causes of the increased prevalence of head lice infestation in Saudi Arabia.

Furthermore, more head lice collected from different places in Saudi Arabia are needed in future research to provide deeper insight into permethrin resistance among Saudi primary school pupils.

### Funding

The authors received no funding for this work.

### Competing Interests

The authors declare there are no competing interests.

### Author Contributions

- Isra M. Alsaady conceived and designed the experiments, performed the experiments, analyzed the data, authored or reviewed drafts of the article, and approved the final draft.
- Sarah Altwaim conceived and designed the experiments, performed the experiments, prepared figures and/or tables, sample collection, and approved the final draft.
- Hattan S Gattan conceived and designed the experiments, performed the experiments, authored or reviewed drafts of the article, and approved the final draft.
- Maimonah Alghanmi analyzed the data, prepared figures and/or tables, and approved the final draft.
- Ayat Zawawi analyzed the data, prepared figures and/or tables, and approved the final draft.
- Hanadi Ahmedah analyzed the data, prepared figures and/or tables, and approved the final draft.
- Majed H Wakid analyzed the data, authored or reviewed drafts of the article, and approved the final draft.
- Esam I Azhar analyzed the data, authored or reviewed drafts of the article, and approved the final draft.

### Ethics

The following information was supplied relating to ethical approvals (i.e., approving body and any reference numbers):

Ethical permission was obtained from the ethical committee of the Faculty of Applied Medical Sciences, King Abdulaziz University (FAM EC2021-10)

## Data Availability

The raw data are available in the Supplemental File.

## Supplemental Information

Supplemental information for this article can be found online at http://dx.doi.org/10.7717/peerj.16273#supplemental-information.

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
