# Peer review of "Prevalence of permethrin-resistant kdr mutation in head lice (Pediculus humanus capitis) from elementary school students in Jeddah, Saudi Arabia"

_PeerJ, doi:10.7717/peerj.16273_

## Round 0.1 · original submission · Major Revisions

The review process is complete, and two thorough reviews from qualified referees are included at the bottom of this letter. Although there is considerable merit in your paper, we identified some concerns that must be considered in your resubmission. Please, provide a better explanation to the study design and description of the criteria to include or exclude subjects to allow inferences from the readers. It is rigorously needed to include the Ethics statement related to the study. The English language also needs to be improved to make the manuscript clear for readers. The references section must be adjusted to meet the PeerJ criteria. The author must dedicate to answer the points raised with the utmost precision and to make all those reconsiderations in the manuscript to be submitted.

Reviewer 1 ·

Basic reporting

The experimental design and the metod didn't provide in detail. Moreover, the ethical concern should be addressed.

Experimental design

It is not clear for the experimental design whther cross-sectional study or prostpective?

Validity of the findings

The finding of this study is valid and may sever as an important information for pyrethroid resistance mutations of head lice in Jeddah, Saudi Arabia.

Additional comments

See in the attacted file.

Annotated reviews are not available for download in order to protect the identity of reviewers who chose to remain anonymous.

Reviewer 2 ·

Basic reporting

The article written in English and clear, unambiguous, technically. Relevant prior literature should be appropriately referenced. Figures are relevant to the content of the article, of sufficient resolution, and appropriately described and labeled.

Experimental design

Original primary research within Aims and Scope of the journal.

Validity of the findings

All underlying data have been provided; they are robust, statistically sound, & controlled. The results are well-stated, linked to the original research question and supporting conclusions.

Additional comments

It is an important study because This is the first study in Saudi Arabia to detect prevalance of permethrin resistance-related mutation in human head lice. The study is well-designed and expressed. All the sections are well-written and easy to understand. The authors have clearly described the materials and methods used in their study. However, reviewing the suggestions I have outlined below may improve the manuscript.
- Both voltage-sensitive sodium channel and voltage-gated sodium channel are used in the manuscript. Using one of them may be more understandable for the reader.
- Line 114-165, These sentences should be reconsidered.
-Line 197, ‘’T929I kdr mutation is one of three mutations found in human head lice’’ this sentence should be revised. Indeed, the T929I kdr mutation has been reported to be associated with permethrin resistance at amino acid sequence positions of the house fly VGSC rather than the head lice amino acid sequence. Maybe it would be more correct to write ’T917I instead of ’T929I.

---

## Round 0.2 · Minor Revisions

The authors addressed the main concerns of the reviews. However, the revised manuscript still deserves attention. Please, provide point-to-point responses according to the comments made the Reviewer #1 in the new version of your manuscript.

Reviewer 1 ·

Basic reporting

The manuscript meets to the standard of the journal.

Experimental design

It's clear for experimental design, but some method doesn't clear.

Validity of the findings

It's necessary to verify the result from the gel electrophoesis by sequencing.

Additional comments

Although the authors have addressed to all inqueries,the sequnce to confirm the gene mutation didn't provide. Moreover, it is not clear how to collect 45 head lice for DNA extraction. One louse was obtained from individual or pooled sample from each individual positive children? What stage of louse was used nymph or adult?

---

## Round 0.3 · accepted · Accept

The authors have satisfactorily addressed all review comments and made the necessary changes to the manuscript.